# Learning poly-synaptic paths with traveling waves

Yoshiki Ito[1]*, Taro Toyoizumi[2,3]*

1 Graduate School of Information and Technology, the Department of Mechano-Informatics, the University of Tokyo, Tokyo, Japan, 2 Lab for Neural Computation and Adaptation, RIKEN Center for Brain Science, Saitama, Japan, 3 Department of Mathematical Informatics, Graduate School of Information Science and Technology, the University of Tokyo, Tokyo, Japan

* y.ito@ne.t.u-tokyo.ac.jp (YI); taro.toyoizumi@riken.jp (TT)

## Abstract

Traveling waves are commonly observed across the brain. While previous studies have suggested the role of traveling waves in learning, the mechanism remains unclear. We adopted a computational approach to investigate the effect of traveling waves on synaptic plasticity. Our results indicate that traveling waves facilitate the learning of poly-synaptic network paths when combined with a reward-dependent local synaptic plasticity rule. We also demonstrate that traveling waves expedite finding the shortest paths and learning nonlinear input/output mapping, such as exclusive or (XOR) function.

## Author summary

There are approximately $10^{11}$ neurons with $10^{14}$ connections in the human brain. Information transmission among neurons in this large network is considered crucial for our behavior. To achieve this, multiple synaptic connections along a poly-synaptic network path must be adjusted coherently during learning. Because the previously proposed reward-dependent synaptic plasticity rule requires coactivation of presynaptic and postsynaptic neurons, learning can fail if a subset of neurons along a distant network path is inactive at the beginning of learning. We suggest that traveling waves that are initiated at an information source can mitigate this problem. We performed computer simulations of spiking neural networks with reward-dependent local synaptic plasticity rules and traveling waves. Our results show that this combination facilitates the learning and refinement of synaptic network paths. We argue that these features are a general biological strategy for maintaining and optimizing our brain function. Our research provides new insights into how complex neural networks in the brain form during learning and memory consolidation.

## Introduction

Waves of neural activity in the brain play an essential role in recognition and learning [1]. Among them, traveling waves are observed at different spatial scales in many brain regions by

**Funding:** Brain/MINDS from Japan Agency for Medical Research and Development [AMED] under

Grant Number JP21dm020700 (T.T.) https://www.
amed.go.jp/en/index.html Japan Society for the
Promotion of Science [JSPS] KAKENHI Grant
Number JP18H05432 (T.T.). https://www.jsps.go.
jp/english/index.html The funders had no role in
study design, data collection and analysis, decision
to publish, or preparation of the manuscript.

**Competing interests:** The authors have declared
that no competing interests exist.

different recording methods, such as electroencephalogram (EEG) [2–4], voltage-sensitive
dyes (VSDs) [5,6], and local field potentials (LFP) [7,8]. Traveling waves are typically observed
under mild anesthesia [7,9], sleep [10], or idle [11].

Cortical traveling waves consist of the upstate and downstate of neurons and propagate
these phases coherently [12–15]. The upstate is defined by relatively large membrane potential
fluctuations with a high firing rate, while the downstate is referred to as a phase of small fluctu-
ations with little spikes [16]. The propagation of this up/down state is estimated to be slower
than the axonal signal transmission, and the activity spreads both as subthreshold and supra-
threshold responses [17]. Lubenov et al. [18] suggested that these traveling waves spread along
with anatomical structures rather than spatial distance.

The role of traveling waves has been unclear. One hypothesis is that traveling waves mediate
lateral propagation of signals within the cortex [7,19]. Rubino et al. [20] suggested that the
waves mediate information transfer to distant neurons during movement preparation and exe-
cution. Another hypothesis is that slow oscillations during sleep contribute to memory consol-
idation [21,22]. Notably, while these works suggest the significance of traveling waves for
learning, specific mechanisms of how traveling waves improve learning are yet to be uncov-
ered. We conducted computer simulations of neural network models to study this.

To explore this mechanism, we modeled synaptic plasticity. Synaptic weight between a pair
of neurons changes according to presynaptic and postsynaptic neural activity and a reward sig-
nal [23–25]. Reward-modulated spike-timing-dependent plasticity (STDP) strengthens synap-
ses that contribute to eliciting a spike in the presence of a reward signal [26,27]. While this
learning rule tends to increase the probability of reproducing a spike sequence that leads to a
reward, it cannot efficiently associate spiking activity among indirectly connected neurons.
Signal transmission between indirectly connected neurons is crucial for task performance [28]
because most neurons in the brain are connected indirectly [29].

We hypothesized that a critical role of traveling waves is to propagate neural activity
between distant and indirectly connected neurons. Consistently, Lubenov et al. [18] reported
that theta waves in the hippocampus assist signal transmission across areas, such as the amyg-
dala, hypothalamus, and medial prefrontal cortex, and this is also suggested in humans
[30,31]. Together with the standard reward-independent STDP [32], traveling waves could
gradually create a repertoire of paths spreading from a wave-initiating site. Once such a reper-
toire is prepared, neurons are coherently activated along the paths so that reward-modulated
STDP could select a subset of these paths to perform a task. We simulate computational mod-
els of reward-modulated STDP to study if traveling waves enhance learning.

## Results

To test our hypothesis, we used relatively small excitatory spiking neural networks ($N \sim 100$)
with a global inhibitory signal and a global dopaminergic signal. Fig 1A explains the scheme of
our setting. For the spiking neuron model, we adopted the leaky integrate-and-fire neuron.
The dynamics of the membrane potential $v_i$ of neuron $i$ are described by

$$\mathrm{d}v_i/\mathrm{d}t = [v_0 - v_i + h_i + h_i^{\mathrm{ext}} - h^{\mathrm{inh}}]/\tau + \sigma_i(t)\xi_i \tag{1}$$

where $v_0$ = -70 mV is the resting potential, $h_i$ is the synaptic input from surrounding excitatory
neurons to neuron $i$, $h_i^{\mathrm{ext}}$ is the external input to neuron $i$, $h^{\mathrm{inh}}$ is an inhibitory feedback signal
that controls the overall firing rate of the network, computed as the running average of spikes
from all neurons (see Material and Methods), and $\tau$ = 10 ms is the membrane time constant. $h_i$
is updated according to $\mathrm{d}h_i/\mathrm{d}t = -h_i/\tau_h + h_0 \sum_j S_{ij} f_j(t - t_d)$, with synaptic time constant $\tau_h$ = 5 ms,
scaling constant $h_0$ = 60 mV, excitatory synaptic weight $S_{ij}$ from neuron $j$ to neuron $i$, spike-

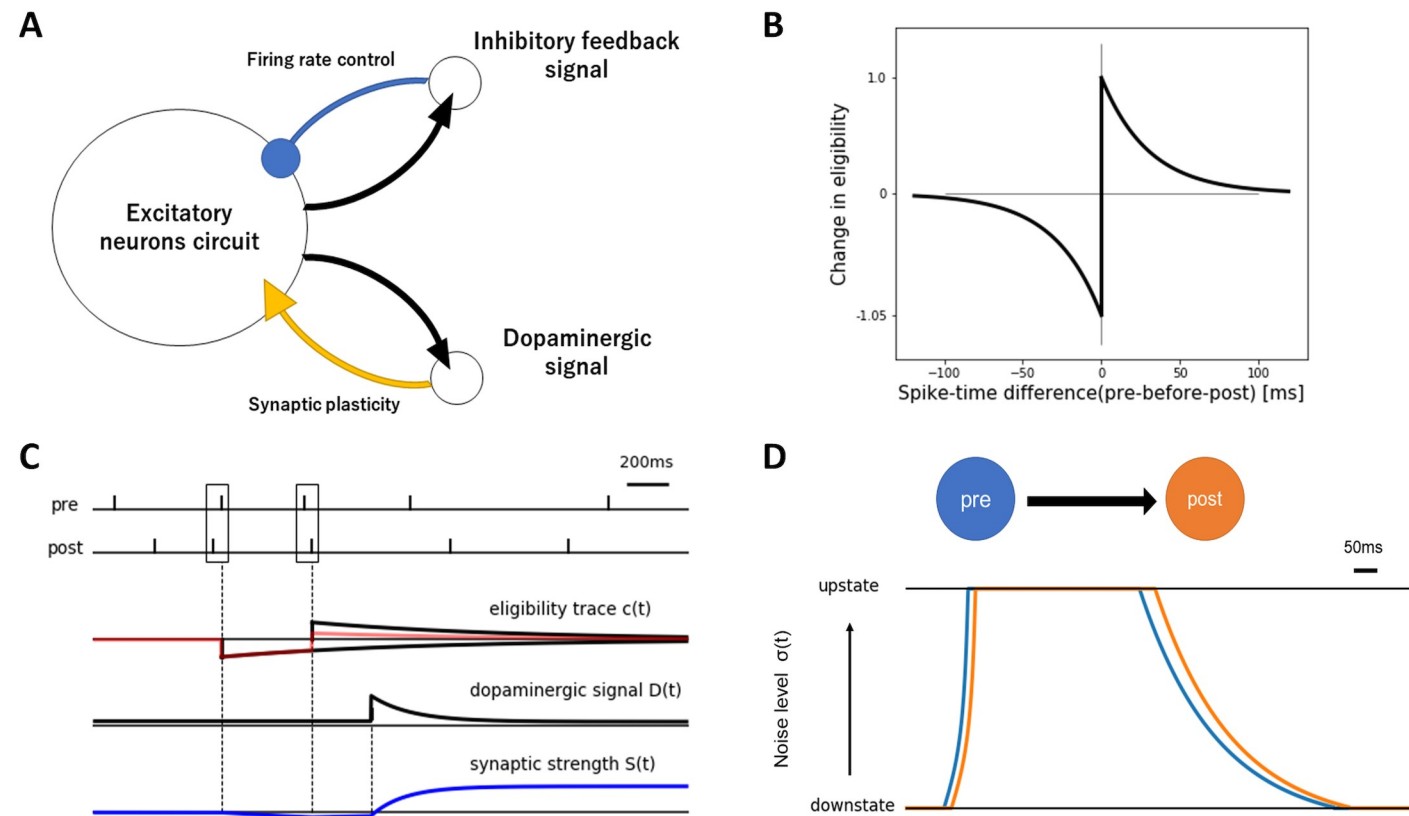

**Fig 1. Schematic explanation of the modified reward-modulated STDP rule.** (A) The whole network overview. (B) The STDP learning window. (C) The mechanism of synaptic plasticity. Synaptic weight changes as a product of eligible trace $c(t)$ and dopaminergic signal $D(t)$. (D) The upstate propagation from a presynaptic neuron to a postsynaptic neuron.

train $f_j$ of neuron $j$ as a sum of delta functions peaking at neuron $j$'s spike timing, and synaptic transmission delay $t_d = 2$ ms. The neuron emits a spike when $v_i$ reaches a spiking threshold of -54 mV and then is reset to resting potential at -60 mV. In addition, each neuron receives uncorrelated white Gaussian noise $\xi_i$. The noise level is controlled by a time-dependent standard deviation $\sigma_i(t)$, modulated by traveling waves as described below. A subset of neurons (stimulated neurons) receives external input as $h_i^{\text{ext}}$ and other neurons receive no external input, $h_i^{\text{ext}} = 0$ mV. The stimulated neurons receive input pulses at 200 Hz as $h_i^{\text{ext}}$ that enforces them to spike during the first 250 ms of each learning trial (see below for each task setup).

As a synaptic plasticity rule (Fig 1B and 1C), we used a modified version of the reward-modulated STDP [26]. In this conventional model, synaptic plasticity does not occur in the absence of reward or punishment. However, recent research suggests that the dopaminergic signal has two different timescales: tonic and phasic [33]. Therefore, we prepared the corresponding tonic variable $D_t$, which represents the baseline dopamine level and the phasic variable $D_p$, which represents the dopaminergic signal driven by a reward or punishment. Hence, we assume that $D_t$ signaling induces reward-independent STDP, and $D_p$ signaling induces reward-dependent STDP. The amount of reward or punishment exponentially declines after the stimulation offset with a decay time-constant of 200 ms. Both dopaminergic components are assumed to be modulated by the novelty [34] of the task. Toward the end of the simulations, both $D_t$ and $D_p$ slowly declined to terminate learning and fix the network (see Material and Methods). Note that dopaminergic signals $D_t$ and $D_p$ are global variables common to all

synapses. The synaptic weight $S_{ij}$ ($0 \leq S_{ij} \leq S_{max}$) from neuron $j$ to $i$ is adjusted when $c_{ij} > 0$ or $D_\mathrm{p} > 0$ according to

$$\mathrm{d}S_{ij}/\mathrm{d}t = c_{ij}(D_\mathrm{t} + D_\mathrm{p})/\tau_s \tag{2}$$

where $S_{max} = 0.24$ is the maximum synaptic weight, $\tau_s = 1$ ms is a time unit, and $c_{ij}$ ($-S_{max}/2 \leq c_{ij} \leq S_{max}/2$) is the so-called STDP eligibility trace [26] that accumulates the effects of plasticity events with time-constant $\tau_c = 1000$ ms, namely,

$$\mathrm{d}c_{ij}/\mathrm{d}t = -c_{ij}/\tau_c + \gamma \cdot (f_i \bar{f}_j - 1.05 \cdot \bar{f}_i f_j) \tag{3}$$

where $f_i$ is the spike-train of neuron $i$, and $\bar{f}_i$ is the running average of $f_i$ with a time constant $\tau_{STDP}$. The increment of $c_{ij}$ follows a typical asymmetric STDP window [35] with amplitude $\gamma = 0.0009$ and time-constant $\tau_{STDP} = 30$ ms (Fig 1B). The $c_{ij}$ instantaneously increases if there is a pre-before-post-event, instantaneously decreases if there is a post-before-pre-event, and otherwise exponentially decays with the time-constant $\tau_c$. The upper and lower bounds of $c_{ij}$ limit the speed of synaptic change. We assumed no changes in the synaptic weight when $c_{ij} < 0$ and $D_\mathrm{p} < 0$.

For the wave, we used a simple custom-made propagation rule. The upstate is defined as a high noise level state ($\sigma_i(t) \sim 6$ mV), while the downstate is a low noise level state ($\sigma_i(t) \sim 3$ mV). These noise levels roughly reproduce the experimentally observed firing rate of 5 Hz in the upstate and 0 Hz in the downstate [36]. The initial upstate spread from externally stimulated neurons in each trial. Then, the upstate propagates from these neurons to the peripheral neurons. The noise level is determined by $\sigma_i(t) = \alpha_i \cdot \psi_i + 3$ mV with influx coefficient $\alpha_i$ (see Material and Methods) and local field $\psi_i$, representing the average activity of a non-modeled neuron mass around the modeled neuron $i$. To control the noise level, we constrained the range of $\sigma_i(t)$ between 3 and 6 mV and the range of $\psi_i$ between -1 mV and 100 mV. $\psi_i$ is updated (Fig 1D) by

$$\begin{aligned}
\mathrm{d}\psi_i/\mathrm{d}t = {} & (g_i(t) - \psi_i)/\tau_w \\
& + \left( \frac{0.2}{\delta t} \sum_{j \to i} [\psi_j(t - \delta t) - \psi_i(t) - \theta]_+ - \frac{0.1}{\delta t} \sum_{j_{i \to}} [\psi_i(t - \delta t) - \psi_j(t) - \theta]_+ \right)
\end{aligned} \tag{4}$$

where $\tau_w = 200$ ms is the time constant of waves, $\delta t = 20$ ms is a propagation delay, $\theta = 0{,}001$ is a threshold for wave propagation, the expressions $j_{\to i}$ and $j_{i \to}$, respectively, represent the sets of $j$ indices that have connections incoming to and outgoing from neuron $i$. $[x]_+$ is the rectified linear function that takes $x$ for positive $x$ and 0 otherwise. $g_i(t)$ describes the time-dependent drive for the local field $\psi_i$ by the external input. For stimulated neuron $i$, $g_i(t) = \eta \int_{t_{\mathrm{on}}}^{t} \delta(\mathrm{mod}(t', 5\mathrm{ms}))dt'$ integrates the external input from stimulation-onset time $t_{\mathrm{on}}$, while time $t$ is in the stimulation interval, where mod is the modulo function. Thus, $g_i(t)$ discontinuously increases by $\eta$ every 5 ms but is constant within this interval. We assume that $g_i(t) = -5$ mV after the stimulation interval. For non-stimulated neurons, $g_i(t) = 0$ mV always holds. The gain factor $\eta$ takes a task-dependent value as described in Material and Methods. Altogether, the local field around stimulated neurons rapidly increases at the beginning of each learning trial and then diffuses as a wave to the local field of connected neurons. By the end of the learning trial of duration 3.0 s, $\psi_i$ for all neurons decay close to zero. Neurons are placed on a two-dimensional square sheet. A rigid boundary condition is used so that the waves collapse at the edges of the sheet. To highlight the role of traveling waves, we also simulate models without waves. A constant noise level, $\sigma_i$, is used in these models. The value of $\sigma_i$ is chosen so that the overall firing rate is the same as that of the

corresponding model with waves. We define the conventional model as the model without the tonic dopamine signal and traveling waves.

Below, we conducted three tasks to illustrate our points. In Task 1, we demonstrate that the combination of reward-dependent STDP and traveling waves can selectively reinforce reward-related paths. In Task 2, we show that traveling waves can empower reward-dependent STDP to reinforce initially weak shortcut paths. In Task 3, we show that the reward-dependent STDP and waves can be exploited to learn the XOR function.

## Task 1: Selectively reinforcing poly-synaptic paths

First, we demonstrate that the combination of traveling waves and the STDP rule can strengthen a specific path from a stimulated neuron to a target neuron. This task is especially important in large-scale networks such as the brain because most neurons are indirectly connected. A local STDP rule alone does not efficiently solve this task because coherent activation of distant neurons is rare before learning. Wave signals compensate for this deficiency and facilitate the learning of poly-synaptic paths. This effect turns out to be evident, especially in the presence of the tonic dopaminergic signal $D_t$, which is not included in the conventional reward-modulated STDP rule. The $D_t$ signal induces a reward-independent STDP that works synergistically with traveling waves to prepare a repertoire of paths starting from the stimulated neuron (see below).

Fig 2 shows the setting and results of this task. Fig 2A Left shows the initial network setting of this task. The central neuron S with coordinates (600 μm, 600 μm) is stimulated by external input. This task aims to strengthen the path from S to target neuron T positions at the bottom (600 μm, 100 μm). We also prepared three false-target neurons F at the left (100 μm, 600 μm), right (1100 μm, 600 μm), and top (600 μm, 1100 μm), respectively. Synaptic connections from S are all outward, while the synaptic connections to T and F are all inward. Other neurons are randomly and unidirectionally connected to adjacent neurons within $200\sqrt{2}$ μm with a probability of 0.5. If a neuron is isolated by chance, we repeat the procedure until it gets connected. We used this recurrently connected neural network to model a two-dimensional cortical sheet. The task we consider is information routing in a cortical sheet required for some animal tasks, such as learning an appropriate action in response to a stimulus by preparing a path from visual neurons to motor neurons [37]. The central neuron was stimulated during the first 250 ms of each trial. This causes a traveling wave to build up there and spread to the surrounding neurons gradually. A reward or punishment signal is provided (see Material and Methods for details) if the summed spike-count from the target or non-target neurons reaches a threshold level of 5 in each trial. If the target neuron spikes more than the other three false-target neurons during and after the stimulation, the reward signal $D_p$ ($> 0$) is provided to the whole network. Meanwhile, if any of the false-target neurons spikes more than the target neuron, the punishment signal $D_p$ ($< 0$) is provided. We repeated this trial of 3.0 s in duration for 80 times.

In a successful case, the paths from the stimulated neuron at the center to the target neuron at the bottom are selectively strengthened (Fig 2A). Fig 2B shows a successful example of the firing rate of the target (red) and the false-target neurons (black). The firing rate of the target neuron was selectively increased. Fig 2C indicates that the correct paths are gradually strengthened. In the last trial, the combination of waves and the $D_t$ signal successfully establishes a path from the stimulated neuron to the target neuron (Fig 2D). The success rate of each condition is indicated in Fig 2E. Our full model shows the best task performance, while the conventional model (without waves and the $D_t$ signal) fails in this task. For this task to be completed, the $D_t$ signal is critical because the input signal from the stimulated neuron does not reach the target neuron in the initial setting (S1 Fig). Hence, a reward or punishment signal is too unreliable to

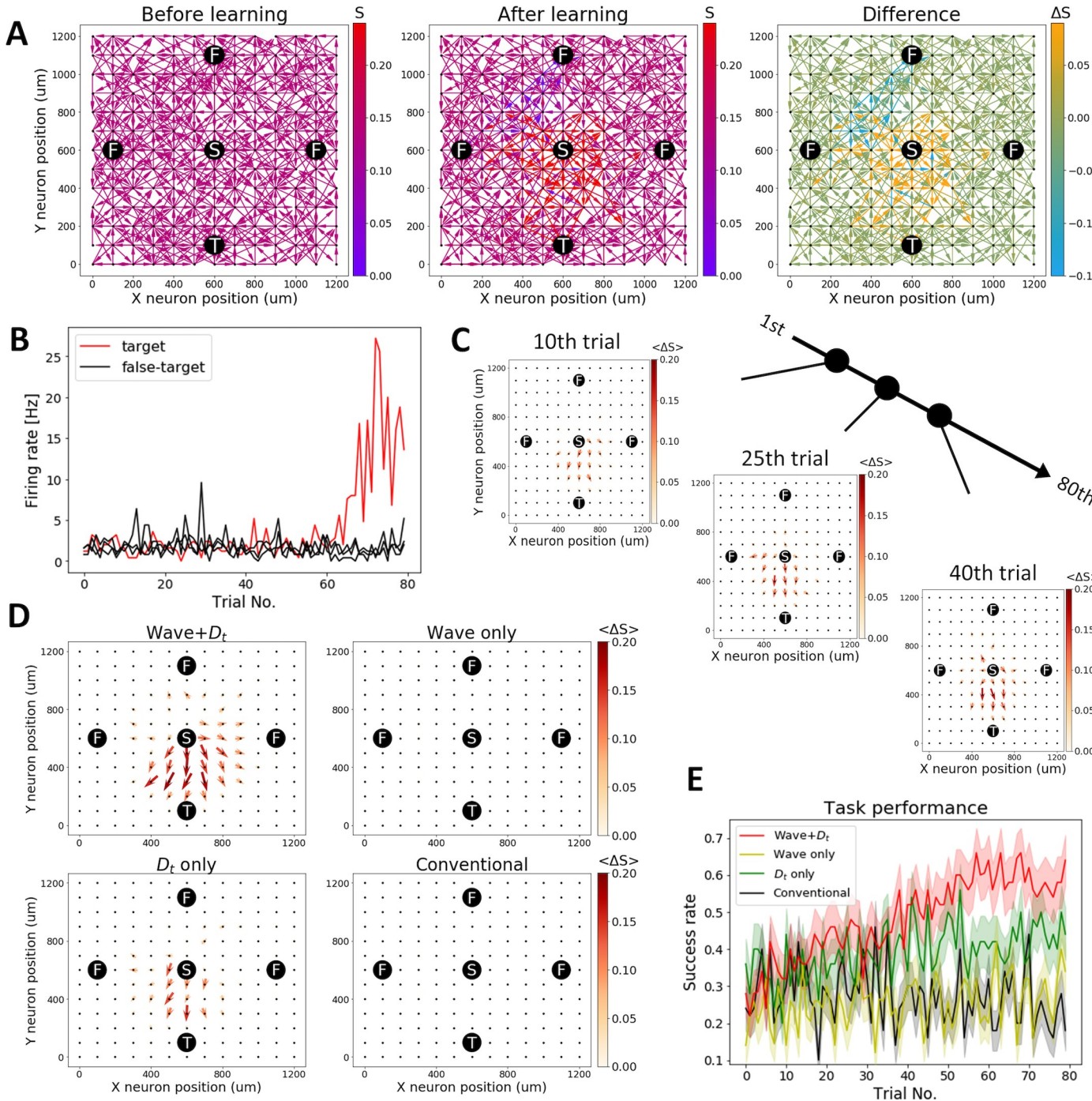

**Fig 2. Upstate propagation improves the reinforcement task of poly-synaptic paths.** (A) An example of a successful trial. The initial synaptic weights are represented in color (Left). All neurons are aligned in a grid with 100 μm spacing, and the adjacent neurons within $200\sqrt{2}$ μm are randomly connected with a probability of 0.5. Synaptic connections from stimulated neuron S are all outward, while the synaptic connections to target neuron T and false-target neurons F are all inward. The path from S to T is selectively strengthened at the end of the learning (Middle). The difference between the initial synaptic weight and the final synaptic weight (Right). (B) A successful example of this task. The firing rate of the target neuron selectively increases. (C) The averaged synaptic weight difference from the initial condition to the $10^{th}$, $25^{th}$, $40^{th}$ trials is plotted. The averaged synaptic weights are calculated, including the direction of synaptic weights (so that the opposite direction has a minus sign). (D) The averaged synaptic weight difference between the initial trial and the last trial (the $80^{th}$ trial) is plotted. (E) The success rate of each condition (50 simulations averaged). The shaded area indicates the standard error of the mean. A combination of wave and tonic dopamine signal $D_t$ (red line) shows the best task performance, while the conventional model (black line) fails to complete this task.

train the network at the beginning. In contrast, reward-independent STDP, induced by the $D_t$ signal, gradually establishes radially symmetric outbound paths spreading from the stimulated neuron (S2 Fig). Traveling waves speed up this process by enhancing radial spreading neural activity, but they are not effective in the absence of the $D_t$ signal because they drive noisy neural activity (S1 Fig). Once radially symmetric candidate paths were formed (S2 Fig), reward-modulated STDP can select paths toward the target neuron based on reward and punishment signals (Fig 2).

## Task 2: Finding a shortcut

The combination of the wave signal and STDP rule can also help find the shortest paths from the stimulated neuron to a target. Generally, finding short paths is vital for fast and reliable computation—transmission through detour paths is slow and fragile because successful transmission depends on multiple neurons' states, which are unreliable in nature. Finding an initially weak shortcut path might be difficult without traveling waves because the neurons along the shortcut path would seldom be activated coherently. Wave propagation can significantly increase this probability and accelerate the learning process.

Fig 3 shows the setting and results of this task. Similar to Task 1, we placed a stimulated neuron and a target neuron. The stimulated neuron S is located upper-left at (100 μm, 500 μm), and the target neuron T is located bottom-left at (100 μm, 100 μm) (Fig 3A). Neurons within $100\sqrt{5}$ μm are randomly and unidirectionally connected with a probability of 0.5. If a neuron is isolated by chance, we repeat the procedure until it gets connected. The synaptic weights of a detour path are initially set three times as strong as the other synapses. The stimulated neuron receives external input at the beginning of each trial for 250 ms. Initially, the signal is only transferred through the detour path, which takes more than 160 ms to reach the target neuron. Meanwhile, it takes less than 100 ms when the signal is transferred through the shortcut paths after learning. This setting could reflect inter-regional signal transmission, for example, where the shorter paths represent direct signal transmission, and the detour paths represent the signal transmission via several relay stations.

In a successful case, shorter paths are strengthened while the detour paths are moderately strengthened (Fig 3A). This network change occurs with a continuous reinforcement of shortcut paths (Fig 3B). The wave condition successfully establishes shortcut paths, while the no-wave condition cannot strengthen them (Fig 3C). The $D_t$ signal enhances the role of waves by further strengthening the shortcut paths by reward-independent STDP but is not effective on its own because synapses along the shortcut paths are initially too weak to induce spiking activity in the absence of waves. Fig 3D shows the overall performance of this task. Note that the amount of reward declines with the latency of activating the target neuron (see Material and Methods). The wave condition with the $D_t$ signal outperforms the conventional model. Fig 3E represents the averaged latency index for obtaining a reward after trial onset. The latency index is equal to the latency of the first spike in the target neuron after the stimulus onset but saturates for latency above 300 ms to be insensitive to outliers. The latency index decreases faster in wave conditions than in no-wave conditions. While the effect of $D_t$ on task performance is evident in these networks of recurrently connected neurons, the effect is less prominent in feedforward networks (S1 Text). This result shows that $D_t$-induced reward-independent STDP is especially important in selectively strengthening outbound paths from the stimulated neuron.

## Task 3: Learning a nonlinear function

In this task, we demonstrate that our model is useful for a more practical setting. Here, we show that the XOR function can be learned in our model as well. Nonlinear functions such as

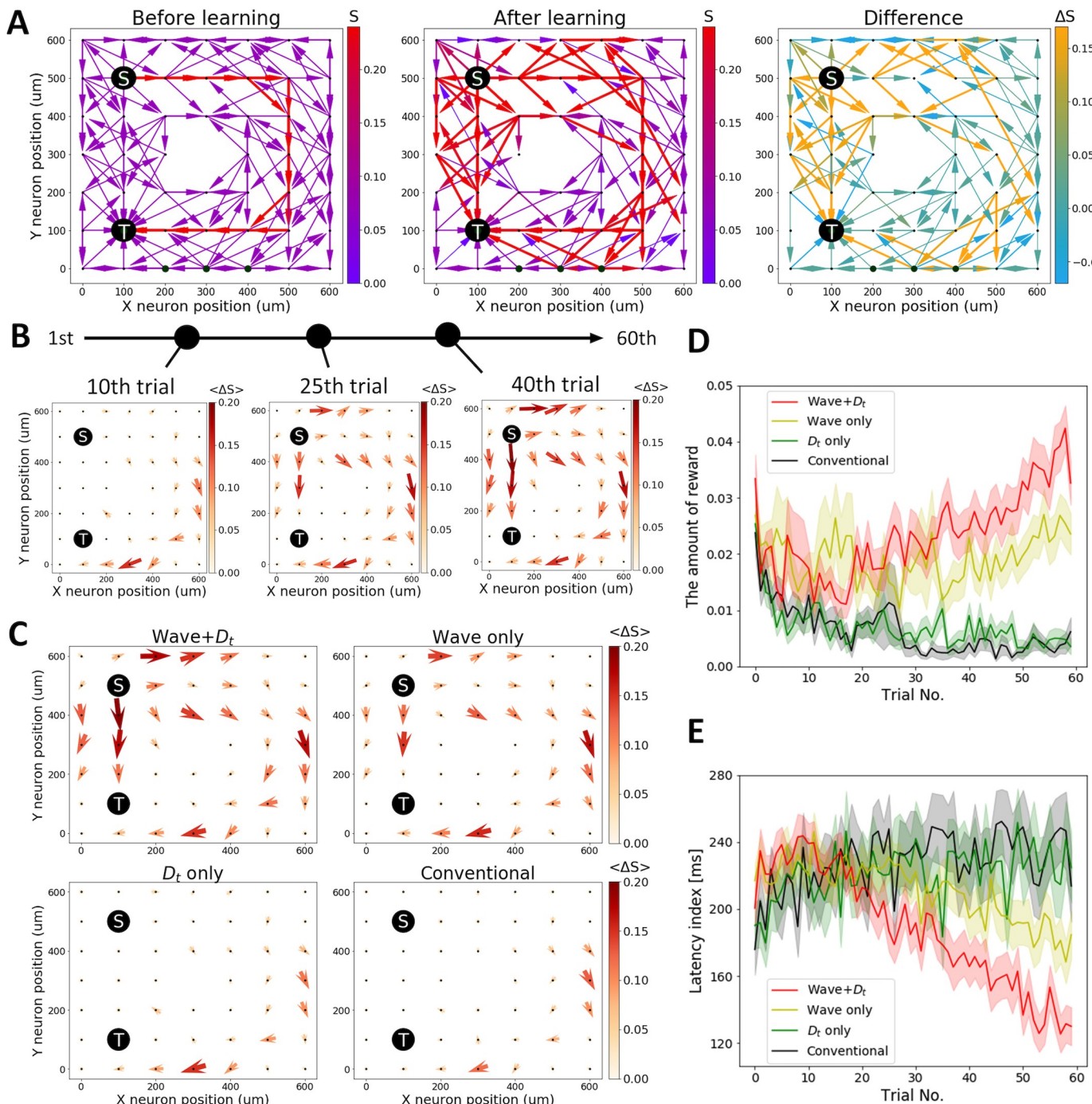

**Fig 3. Wave propagation helps find a shortcut.** (A) An example of a successful trial. Each panel represents the initial synaptic weight (Left), the last synaptic weight (Middle) and the difference between them (Right). Initially, a strong detour path from the stimulated neuron S on the upper-left at (100 μm, 500 μm) to the target neuron T on the bottom-left at T (100 μm, 100 μm) is prepared. Neurons within $100\sqrt{5}$ μm are randomly connected with a probability of 0.5. Synaptic connections from S are all outward, while synaptic connections to T are all inward. At the end of the trial, the shortcut paths are strengthened while the detour paths are preserved. (B) The averaged synaptic weight difference from the initial trial to the 10th, 25th, and 40th trials is plotted. The averaged synaptic weights are calculated by each neuron, including the direction of synaptic connection. (C) The averaged synaptic weight difference from the initial trial and the last trial (the 60th trial) is plotted. The wave conditions can successfully escape a local solution state and reach a better solution for this task. The error bar indicates the standard error of the mean. (D) The amount of reward signal is plotted. The wave conditions can successfully escape a local solution state and reach a better solution for this task. The error bar indicates the standard error of the mean. (E) The latency index takes the latency of the first spike in the target neuron after the stimulus onset if it is below 300 ms and takes 300 ms if the latency is above 300 ms. The condition with waves and tonic dopaminergic signal $D_t$ (red) shows the best performance, while the conventional model (black) fails. The error bar indicates the standard error of the mean.

the XOR function are essential for complex calculation, but how to realize them efficiently with the reward-modulated STDP rule remains to be seen. We propose that our model has an advantage in this task because some nonlinear functions can be created by finding appropriate poly-synaptic paths. Among the various kinds of nonlinear functions, we chose the XOR function because of its simplicity and universality of logic gates [38]. It is widely known that implementing an XOR function requires a hidden layer in a feedforward neural network. Therefore, this task is difficult for the STDP rule because indirect paths should be learned. Our model can alleviate this difficulty and facilitate the learning process.

Fig 4 shows the setting and results of this experiment. In this task, we used four stimulated neurons located at the bottom, namely $0_a$ (15 μm, 0 μm), $1_a$ (45 μm, 0 μm), $0_b$ (75 μm, 0 μm), and $1_b$ (105 μm, 0 μm) (Fig 4A). In the middle line at Y = 100 μm, 120 neurons were aligned. In the initial setting, these middle layer neurons receive a strong projection ($S_{ij}$ = 0.2) from the nearest stimulated neuron and a weak projection ($S_{ij}$ = 0.1) from another randomly selected stimulated neuron. Two target neurons are positioned at the top, namely, F (30 μm, 200 μm) and T (90 μm, 200 μm). Each middle layer neuron has a strong projection ($S_{ij}$ = 0.2) to one of them. During this task, four different stimuli are provided, where one of the pairs of stimulated neurons $0_a0_b$, $0_a1_b$, $1_a0_b$, or $1_a1_b$ receives external input. At the beginning of each trial, the corresponding neurons were stimulated for 250 ms. The target neuron for each of the four stimuli was F, T, T, and F, respectively. If the corresponding target neuron fires more than the other neuron, the reward signal $D_p$ ($> 0$) is provided. Otherwise, the punishment signal $D_p$ ($< 0$) is provided. The reason for initially having weak inputs from the stimulated neurons to the middle layer neurons is to expedite learning. If these connections are strong enough, the task can be solvable simply by learning the output-layer synapses. We set these synapses weak enough so that the task performance remains near the chance level by learning only the output-layer synapses. We use a feedforward network in this task, which may be implemented, for example, in three information-processing layers (e.g., layer 4 to layer 2–3 to layer 5) in a cortical column [11,39–41].

Fig 4A shows the synaptic weight change in a successful case. The relevant connections are selectively strengthened or weakened. Each synaptic path strength is calculated in Fig 4B. Correct paths are gradually strengthened through the trial. In the last trial (the 25[th] trial), the wave condition successfully established the correct paths, while the no-wave condition failed (Fig 4C). Fig 4D shows the task performance for each condition. The wave conditions (red and yellow) perform better than the no-wave conditions (green and black) because, similar to Task 2, the weak connections can only be strengthened with the support of traveling waves. However, the contribution of $D_t$ is small in this task because the signal transmission from the stimulated neurons to the target neurons is easily achieved from the beginning in the presence of waves due to the disynaptic feedforward structure.

## Discussion

We have demonstrated that the combination of traveling waves and tonic dopaminergic signals enhances selective reinforcement of poly-synaptic paths. Further, we showed that this combination is also helpful for learning a shortcut and a nonlinear function. The advantage of traveling waves to send signals across distant neurons is effectively utilized in the tasks we explored. Thus, we argue that a possible role of traveling waves in the brain is to aid local learning rules, such as the reward-modulated STDP, to efficiently learn poly-synaptic paths by inducing coherent activity in neurons along with them.

The advantage of the proposed model over the conventional model is twofold. First, the combination of traveling waves and the tonic dopaminergic signal helps to prepare paths

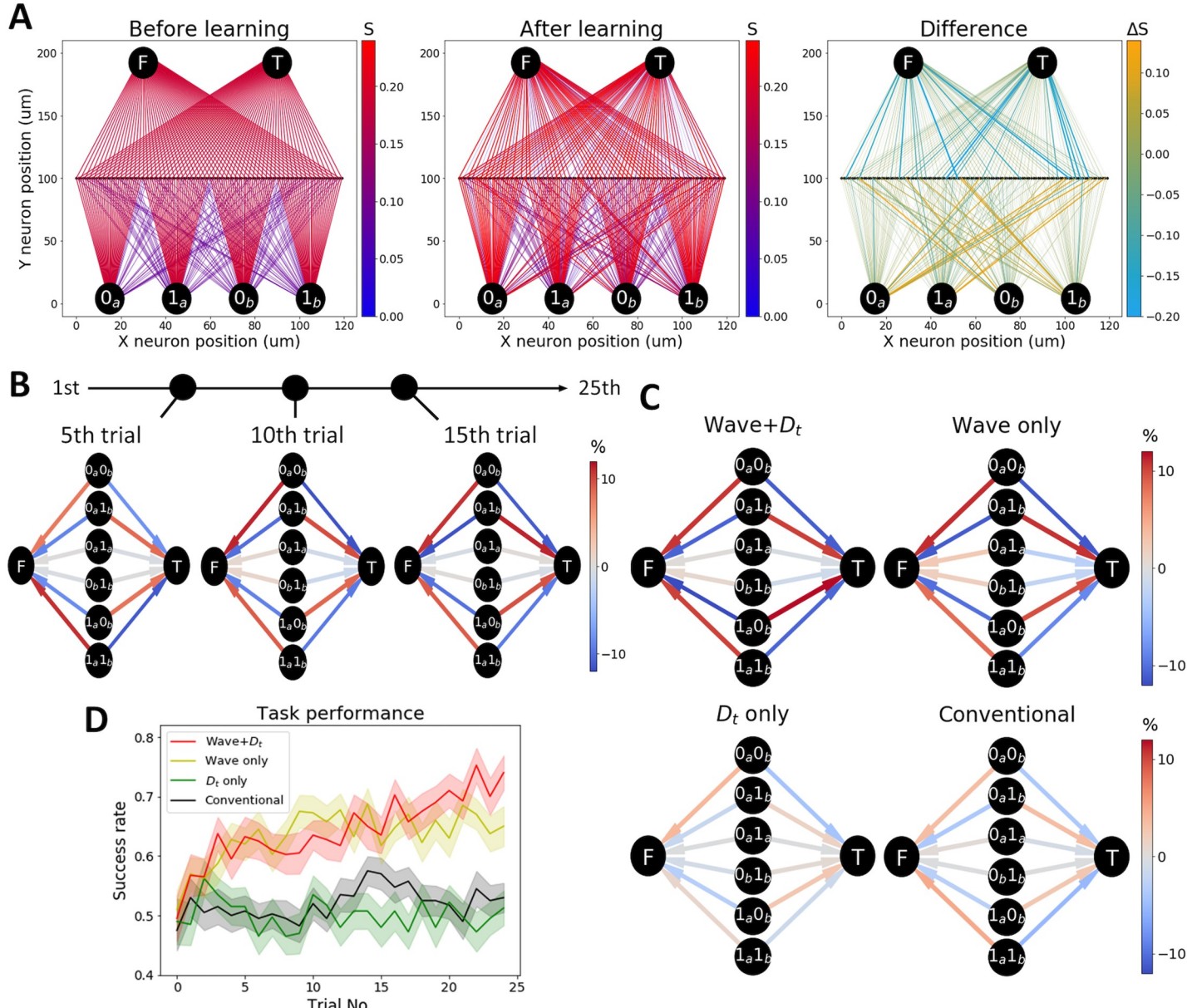

**Fig 4. Wave propagation is useful for learning a nonlinear function.** (A) A successful example of synaptic weight change. The Initial condition (Left), the last condition (Middle), and the difference (Right). Each middle layer neuron receives a strong synaptic weight and a weak synaptic weight from stimulated neurons and sends an output to T or F. The success rate is initially at the chance level. (B) The averaged synaptic weight difference between each middle layer neuron projecting to T and F is plotted at the 5th, 10th, and 15th trials. Each path strength is calculated as a product of the averaged synaptic weight from stimulated neurons to middle layer neurons and middle layer neurons to a target neuron. Percentage changes in averaged synaptic weight are shown in color. (C) The same plots as (B) at the last trial (the 25th trial). Wave condition (top column) successfully learns the correct paths, while no-wave condition (lower column) fails. (D) The success rate of the XOR task. Wave conditions (red & yellow) shows better results than no-wave conditions (green & black).

starting from stimulated neurons. In our model, a tonic dopaminergic signal permits reward-independent STDP. In its presence, traveling waves efficiently create a repertoire of poly-synaptic paths spreading from the wave-initiation sites. Second, once a repertoire of paths from the stimulated neurons is prepared, a reward-dependent phasic dopaminergic signal can reinforce its subset. These features are consistent with the biological evidence of recent studies. Beeler et al. [42] showed that tonic and phasic dopamine have different roles; tonic dopamine

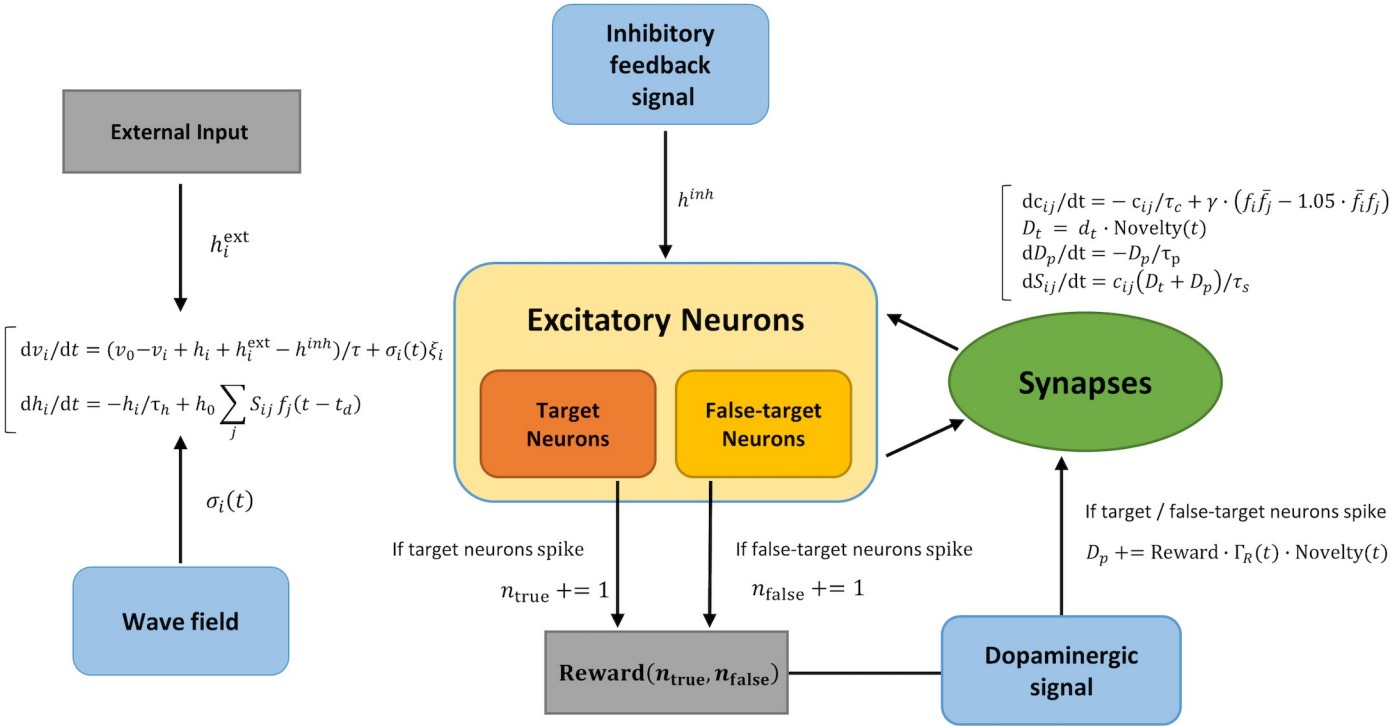

**Fig 5. The whole system of our model.** Excitatory neurons are locally connected via synapses. Inhibitory feedback signal controls the firing rate of excitatory neurons, global dopaminergic signal modulates the synaptic weights, and wavefield created by the activities of other neurons controls the activity level of each excitatory neuron. External input and reward functions are externally provided. The + = operator means that the right-hand-side is added to the left-hand-side when an event happens (with delay for $D_p$). We use the same parameters (except three parameters summarized in Table 1) to learn three qualitatively different tasks in different network architectures, which underscores the robustness of the learning rule and the role of traveling waves.

modulates the degree of learning and its expression, while phasic dopamine is the main source of reinforcement learning. In addition, Schultz [43] suggests that the continuous emission of tonic dopaminergic signals controls the motivation for exploration, while the discrete phasic dopaminergic signal induces event-related synaptic plasticity. Our model is also testable by examining the relationship between traveling waves and learning in a specific environment, such as by selective blockade or enhancement of either the tonic or phasic component of the dopaminergic signal.

Our model suggests a mechanism of memory consolidation during slow-wave sleep. Some experiments have observed traveling waves across the entire brain during slow-wave sleep [9,10] and showed their importance in memory consolidation [21,22]. Importantly, dopaminergic neurons emit tonic signals during slow-wave sleep [44]. These studies indicate that the combination of traveling waves and tonic dopaminergic signals may consolidate memory. Our results agree with this view, supporting that the coherent activation of neurons caused by traveling waves can prepare poly-synaptic paths for more rapid and reliable signal transmission (cf. Fig 3). Further studies on the role of traveling waves and dopaminergic signals on the efficacy of poly-synaptic paths during slow-wave sleep likely elucidate the mechanism of memory consolidation.

One limitation of our model is the separation of dynamics between neural activity and wave propagation. In our model, wave propagation is modeled by the local field without specific relation to the membrane potential of neurons. While this approach is reasonable in our study that involves only a small number of neurons, the local field must be defined by the

average activity of many neurons in reality [45]. Thus, future large-scale simulations could model the relationship between traveling waves and the membrane potential of neurons in an explicit manner. Further, the current model only involves global inhibition, but different classes of inhibitory neurons contribute to up- and down-states in distinct ways [46]. More subtle features of traveling waves might arise from such detailed modeling. Despite these limitations, our simple model revealed a synergy of traveling waves and dopaminergic signals to efficiently learn the directionality of information flow and distant neural network paths in a reinforcement task. This mechanism would be progressively more important for animals with a larger brain because distant and indirect paths are more dominant. Our study underscores the importance of coherent neural activity in the form of waves for coherent learning beyond pairs of neurons.

## Material and methods

### Simulation environment

We conducted all simulations using the Brian2 simulator (https://brian2.readthedocs.io/en/stable/). This is an open Python library that focuses on simulating spiking neurons [47]. The post-analysis of the simulation is performed by custom-made Python code. The source code is provided in S1 File.

### Networks

The network of excitatory neurons is defined task by task (see Figs 2A, 3A and 4A). As described in the Results section, all excitatory neurons receive an inhibitory feedback signal and a dopaminergic signal for simplicity (Fig 1A). The whole system of our model is indicated in Fig 5.

### Inhibitory feedback

The inhibitory feedback signal $h^{\mathrm{inh}}$ controls the firing rate of excitatory neurons. The dynamics of $h^{\mathrm{inh}}$ are described by

$$\mathrm{d}h^{\mathrm{inh}}(t)/\mathrm{d}t = -h^{\mathrm{inh}}(t)/\tau_{\mathrm{inh}} + \beta\sum f_i(t - t_h) \tag{5}$$

where $\tau_{\mathrm{inh}} = 5$ ms is the inhibitory time-constant, $f_i$ is the spike-train (i.e., the sum of delta functions peaking at each spike timing) of neuron $i$, $t_h = 1$ ms is the transmission delay, and $\beta$ is the inhibitory feedback strength. The values of $\beta$, summarized in Table 1, depending on each task because of the difference in the number of neurons and the network structure.

Inhibitory feedback strength $\beta$ roughly correlates with the number of neuron $N$. Dopamine signal initial amplitude $d_{\mathrm{p}}$ is chosen for the best result for each task. Wave amplitude constant $\eta$ is chosen depending on the network structure. Recurrent networks need relatively larger value than feedforward networks (see Supporting Information for solving Task 2 in

**Table 1. The task-dependent variables are summarized.**

|  | Task 1 | Task 2 | Task 3 |
|---|---|---|---|
| N | 136 | 48 | 126 |
| Network type | Recurrent | Recurrent | Feedforward |
| $\beta$ [mV] | 0.13 | 0.05 | 0.13 |
| $d_{\mathrm{p}}$ | 0.01 | 0.005 | 0.002 |
| $\eta$ [mV] | 5 | 5 | 2 |

feedforward networks). Note that, among several parameters in the model, $\beta$, $d_p$, and $\eta$ are chosen as representative parameters that control the basic ingredients in the model: global inhibitory signal, dopamine signal, and traveling waves, respectively.

## Dopaminergic signals

The tonic dopaminergic signal $D_t$ and the phasic dopaminergic signal $D_p$ are essential components of our simulations. The $D_t$ signal is expressed as

$$D_t = d_t \cdot \text{Novelty}(t) \tag{6}$$

with tonic dopamine constant $d_t$ = 0.003, and the novelty function $\text{Novelty}(t)$ (explained below). This setting is fixed in every task we conducted here. We set $D_t = 0$ for the conventional model.

$D_p$ signal ($-0.3 \leq D_p \leq 0.3$) is adjusted depending on the performance of each task. $D_p$ depends on three variables, reward $R$, amplitude function $\Gamma_R$ for the reward, and novelty variable Novelty (Fig 5). $D_p$ exponentially decays according to

$$dD_p(t)/dt = -D_p(t)/\tau_p \tag{7}$$

except when a target/false-target neuron spikes. When a target/false-target neuron spikes at time $t$, $D_p$ instantaneously jumps at time $t+t_p$ according to

$$D_p(t + t_p) + = R(t) \cdot \Gamma_R(t) \cdot \text{Novelty}(t) \tag{8}$$

where decay constant $\tau_p$ = 200 ms and transmission delay $t_p$ = 100 ms. Note that the + = operator indicates that the right-hand side is added to the left-hand-side variable upon a spiking event (with delay $t_p$). We measured the spike counts of target and false-target neurons by vectors $n_{\text{true}}$ and $n_{\text{false}}$, respectively, in each trial (these vectors are reset to zero at the end of each trial). The raw reward $R$ is a function of $n_{\text{true}}$ and $n_{\text{false}}$. For Task 1, we set $R = 0$ when these neurons are not very active, namely, when the total spike-count of one target and three false-target neurons is less than 5. This adds robustness to the simulation results. Once the total spike-count reached 5, $R = 1.0$, when the target spike-count was the greatest and $R = -0.5$ the target spike-count was not the greatest among the four neurons. Therefore,

$$R = (-0.5 + 1.5\, I[n_{\text{true}} > \max(n_{\text{false}})])I[n_{\text{true}} + \text{sum}(n_{\text{false}}) > 5] \tag{9}$$

where $I[\cdot]$ is the indicator function that takes 1 if the argument is true and 0 otherwise. We mean by $\max(n_{\text{false}})$ and $\text{sum}(n_{\text{false}})$ the maximum and the sum of the spike counts of the three false-target neurons, respectively.

For Task 2, there is one target neuron and no false-target neuron. Therefore, we used

$$R = I[n_{\text{true}} > 0] \tag{10}$$

In this task, the punishment ($R < 0$) is not given.

For Task 3, we again considered one target neuron and one false-target neuron. $R = 1$ when the target neuron fires at least more than 5 spikes than the false-target neuron; $R = -1$ when the false-target neuron fires at least more than 5 spikes than the target neuron; and $R = 0$ otherwise. Namely,

$$R = I[n_{\text{true}} \geq n_{\text{false}} + 5] - I[n_{\text{false}} \geq n_{\text{true}} + 5] \tag{11}$$

We set this margin of 5 spikes to induce a clear difference in the number of spikes between the target and false-target neurons.

Next, we introduce the reward-amplitude function $\Gamma_R$. The amount of reward begins to take a non-zero value after the stimulus onset time $t_{on}$, stays fixed until the stimulus offset time $t_{\text{off}}$, and then decays exponentially. Namely,

$$\Gamma_R(t) = d_p I[t > t_{on}]e^{-\frac{t-t_{\text{off}}}{\tau_d}} \qquad (12)$$

with a dopamine decay constant $\tau_d = 200$ ms and the initial amplitude $d_p$, which is set depending on the task (see Table 1).

Finally, we assume that dopamine release increases with novelty [48] and novelty becomes high when the prediction error is high. We simply assume that Novelty ($0 \leq$ Novelty $\leq 1$) decreases by 0.2 at the end of a correct trial and increases by 0.2 at the end of a wrong trial. Here, we introduce task-dependent correct and incorrect criteria. In Tasks 1 and 3, we used $R > 0$ and $R \leq 0$ at the end of each trial to define a correct and incorrect trial, respectively. In Task 2, we used the latency of signal transmission from the stimulated neuron to the target neuron for the criteria. Latency of less than 100 ms is defined as a success.

## Local field and influx coefficients

For the wave, we used a simple custom-made propagation rule. The upstate is defined as a high noise level state ($\sigma_i(t) \sim 6$ mV), while the downstate is a low noise phase ($\sigma_i(t) \sim 3$ mV). The noise level is determined by $\sigma_i(t) = \alpha_i \cdot \psi_i + 3$ mV with an influx coefficient $\alpha_i$ and local field $\psi_i$. The local field is updated as explained in the Results by

$$\begin{aligned} \mathrm{d}\psi_i/\mathrm{d}t = (g_i(t) - \psi_i)/\tau_w \\ + \left( \frac{0.2}{\delta t} \sum_{j \to i} [\psi_j(t-\delta t) - \psi_i(t) - \theta]_+ - \frac{0.1}{\delta t} \sum_{j_{i \to}} [\psi_i(t-\delta t) - \psi_j(t) - \theta]_+ \right) \end{aligned} \qquad (13)$$

As an initial condition, we choose $\psi_i = 0$ for all $i$, which corresponds to the downstate. We assume that upstate is induced by external stimuli (e.g., [49]). The influx coefficient $\alpha_i$ quantifies the sensitivity of neuron $i$'s noise level to $\psi_i$ and is defined by

$$\alpha_i(t) = 5 \cdot \tanh\left( \int_{t_{\text{on}}}^t \sum_{j \to i} \delta(\psi_j(t-\delta t) - \psi_i(t) - \theta) dt \right) \qquad (14)$$

where $t_{\text{on}}$ is again the trial onset. The coefficient $\alpha_i$ counts the number of neighboring local fields that influenced $\psi_i$ in each trial up to time $t$. The tangent hyperbolic function is introduced to implement a saturation effect. For a conventional setting, $\sigma_i(t)$ is set as a constant value adjusted to the same firing rate as the wave condition.

## Supporting information

**S1 Fig. The difference of the signal-driven spikes and traveling-wave-driven spikes of the target neuron in Task 1.** An example of the membrane potential before learning (Top) and the membrane potential of the same neuron after learning (Middle). The red line indicates spike timing. Before learning, the signal from the stimulated neuron does not reach the target neuron and the target neuron does not fire. In contrast, after learning, the external input reaches the target neuron, and the firing rate increases during the stimulus period. Meanwhile, the firing rate of spontaneous spikes driven by traveling wave does not change before and after learning. The noise level (black line) is changed by a traveling wave of upstate (Bottom). During the stimulus period at the onset of a trial, external input (green bar) is provided to the stimulated neuron.
(TIFF)

**S2 Fig. Task performance without the Dp signal in Task 1 and 2.** (A) The contribution of reward-independent STDP is shown for Task 1 by setting $D$p = 0. The average synaptic weights are computed over 40 simulations, and their differences (from the initial trial to the 15th trial) are plotted with the $D$t signaling and traveling waves (Left) and with the $D$t signaling alone (Right). Outbound synaptic weights near the stimulated neuron are strengthened by the $D$t signaling alone but more strongly with waves. In this task, initial synaptic weights are set rather strong. Hence, the stimulated neuron can propagate its activity to neighboring neurons from the beginning, and poly-synaptic paths toward the target and false-target neurons are gradually extended by reward-independent STDP. This happens even without waves but more efficiently with waves that contribute to the outbound spreading of neural activity. (B) The contribution of reward-independent STDP is shown for Task 2 by setting $D$p = 0. The differences of averaged weights (from the initial trial to the 40th trial) are plotted with the $D$t signaling and traveling waves (Left) and with the $D$t signaling alone (Right). The detour path is efficiently strengthened in both cases because it is strong enough to propagate neural activity from the beginning. However, the shortcut path is strengthened only with waves because it is too weak to propagate neural activity at the beginning. Hence, the shortcut path requires waves to propagate neural activity only with waves, which is required to gradually strengthen the path by reward-independent STDP.
(TIFF)

**S1 Text. Task 2 with feedforward network.**
(PDF)

**S1 File. The source code used for simulations.**
(GZ)

## Author Contributions

**Conceptualization:** Yoshiki Ito, Taro Toyoizumi.

**Data curation:** Yoshiki Ito.

**Formal analysis:** Yoshiki Ito, Taro Toyoizumi.

**Funding acquisition:** Taro Toyoizumi.

**Investigation:** Yoshiki Ito.

**Methodology:** Yoshiki Ito.

**Project administration:** Taro Toyoizumi.

**Resources:** Yoshiki Ito, Taro Toyoizumi.

**Software:** Yoshiki Ito.

**Supervision:** Taro Toyoizumi.

**Validation:** Yoshiki Ito.

**Visualization:** Yoshiki Ito.

**Writing – original draft:** Yoshiki Ito, Taro Toyoizumi.

**Writing – review & editing:** Yoshiki Ito, Taro Toyoizumi.

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
