## [Decision Letter · Decision Letter 0]

30 Mar 2020

Dear Mr. Ito,

Thank you very much for submitting your manuscript "Learning distant paths with traveling waves" for consideration at PLOS Computational Biology.

As with all papers reviewed by the journal, your manuscript was reviewed by members of the editorial board and by several independent reviewers. In light of the reviews (below this email), we would like to invite the resubmission of a significantly-revised version that takes into account the reviewers' comments.

We cannot make any decision about publication until we have seen the revised manuscript and your response to the reviewers' comments. Your revised manuscript is also likely to be sent to reviewers for further evaluation.

Sincerely,

Brent Doiron

Associate Editor

PLOS Computational Biology

Kim Blackwell

Deputy Editor

PLOS Computational Biology

Reviewer's Responses to Questions

**Comments to the Authors:**

Reviewer #1: The authors have presented modeling work that supports the hypothesis that waves in neuronal tissue facilitate memory consolidation. Their simple network model combines reinforcement learning through two-component dopamine modulation, local STDP, and excitatory waves of activity of a mean field coupled to local integrate-and-fire neurons. They show, through three different example tasks, that the waves depolarize broad swathes of neurons, allowing for better polysynaptic pathways to be found and reinforced through STDP and both tonic and phasic global dopamine modulation. Their model builds off of prior theoretical work that combined STDP and reinforcement learning using a single dopamine component.

I think this study is novel and of significant biological interest, uniting three different phenomena under the umbrellas of memory consolidation and computational efficiency. My main concern is about the setup of the tasks that the authors might be able to address in a revision:

1) I was a bit unsure as to the biological motivation and context for the different network architectures. In contrast, the most direct preceding work (as referenced by the authors, Izhikevich 2007) studies a fairly realistic setup involving a randomly-connected sparse recurrent network with 80% excitatory and 20% inhibitory neurons representing L 2/3 in cortex. As it stands, in the present work there are three different architectures that require three different sets of parameters, including numbers of neurons, to perform the tasks well, and no suggestion as to the biological circuits in which they might be instantiated.

1a) The first, recurrent, network, which seems to have all-to-all nearest-neighbor connectivity for the non-stimulated and non-target neurons, seems the most plausible of the three. Should this be considered a part of a recurrent cortical layer as in Izhikevich?

1b) The second network is a simpler feedforward network. While feedforward networks may approximate the architecture in a cortical column or between different areas, it isn’t clear if this second network matches these two contexts. Would it be possible to carry out this task within the recurrent network from task 1? If not, perhaps this can be mentioned as a current limitation, and suggested paths forward in this direction could be touched on in the discussion?

1c) The third network is a highly structured feedforward network. Given the spatial scale, should we consider this to be an approximation of elements from three different layers of a cortical column? If not, providing a specific setting in which such a network might be instantiated would help motivate the biological plausibility of Task 3.

I believe the study, while interesting and novel as it stands, would be strengthened by addressing these points above.

My more minor suggestions are:

i) While the results provided in the plots were described well and concisely, it would be nice to read a fuller examination that might suggest at least some further heuristic explanations for the results. Discussing these differences outlined below and what might account for them will help the reader to gain further insight into the author’s model:

ia) All three tasks show the “conventional” setup at or tied for the bottom of the four combinations tested. However, in Task 1, Dt was apparently an important isolated factor while waves seemingly lead to no increased success on their own, whereas in Tasks 2 and 3, waves seemed to contribute almost entirely to the increased success while Dt was far less important. Indeed, in Task 3, it is unclear if Dt helped at all, whether in isolation or in combination with waves, whereas it clearly provides a boost in Task 2.

ib) Less importantly, while waves do better eventually in Task 2, they seem to do worse for the first 15 trials. Is there a reason it takes this long for them to do well (this is not the case in Task 3)? Is there an obvious reason that the “conventional” condition increases its success rate very rapidly, but then stays essentially flat thereafter?

ii) The “conventional model” does not seem to ever be defined, instead referring the reader to Izhikevich 2007 on line 200. I take it that it’s simply Dp with no Dt and no waves, in which case it would increase readability to define it here. It’s also a little unclear if this combination of Dp with no Dt and no waves can be so directly compared to the model in Izhikevich 2007 as to call it the “conventional model", given the large differences in architecture and model framework (e.g., all-to-all vs sparse connectivity, global inhibition vs. 20% inhibitory cells, ~100 neurons vs ~1,000).

iii) It would be helpful to explicitly define each network architecture and their initial conditions (initial weights), rather than just relying on the reader to inspect the graphs (which are a bit fuzzy). E.g., apart from the stimulated and target neurons for Task 1, is it a nearest-neighbor all-to-all network? Does the stimulated neuron connect only outwardly to all of its nearest neighbors (i.e., a nearest-neighbor source), and do the target neurons connect only inwardly to all of their nearest neighbors (i.e., nearest-neighbor sinks)?

iv) It’s nice to see the examples in Figs 2A, 3A, and 4A. However, since the main results are averages over many realizations, might it also be helpful to show, perhaps in a middle row, the average of these realizations? Perhaps the averaged graph patterns will provide some additional useful perspective and intuition?

v) Especially for Task 1, it seems there is substantial variation in the average performances (e.g., Fig 3C). It would be nice to have some handle on this variation, perhaps by providing shaded regions over the averages that represent the standard error over simulations for each condition?

vi) Some additional description of Task 3 might be helpful for biologists without much of a CS background. A simple definition of XOR would be helpful, along with a description of how your task maps to this nonlinear function. To that end, some more intuitive neuron labels might be helpful. E.g., change (A,B,C,D) to (0_a, 0_b, 1_a, 1_b), and change (P, Q) to (T, F). In that case, (0_a, 1_a) and (0_b, 1_b) both map to T, and (0_a, 0_b) and (1_a, 1_b) both map to F, as one should expect for XOR.

vii) I’m a little confused by the inset in Fig 3C. It says that the plots represent averages over successful trials (line 249), and most values are above 100 ms. But line 230 defines a successful trial as one wherein the latency is less than 75 ms. Given the main plot of Fig 3C, it seems that this definition is a trial-to-trial metric. Perhaps I’ve missed something? Also, is there a reason that "Dt only" and “conventional”, which also have a nonzero success rate after trial 0, are not included in the inset?

viii) I think that the presentation of how the waves are generated would be easier to read if the mathematical formulation for the waves given by \\psi_i (line 150) were presented at the same time as that of \\alpha_i in Methods, in part because they have shared parameters that could be described all at once. Overall, I think readability would increase greatly if lines 147 - 160 were moved to Methods, where \\alpha_i is described in detail (i.e., from the sentence that begins at the end of line 147 to the sentence that ends at the end of line 160).

ix) Dp, while more complicated, would be easier to digest quickly if its description had the same format as Dp and the other main variable presentations. Instead of describing it as a product of items in line 431 that were first explained, readability would be improved if it were first introduced in equation form, like the other main variables, on line 387, proceeded by descriptions of its constituent variables.

Reviewer #2: Review summary for the article titled “Learning distant paths with traveling waves”

The authors have adopted a reward-based learning rule to learn distant paths between a stimulated and a target neuron in a spiking network. This mechanism involves traveling waves to facilitate communication between neurons and to find the shortest path between neurons to propagate the input stimulus. In three different tasks, they have shown an improvement compared to the model suggested in ref 26. However, there are several issues with the current report:

Major points:

1- The novelty of the work is not clear. Is it the involvement of the traveling wave in weight updates through the eligibility trace between neurons? The mechanism has not been clearly highlighted, and it is not comprehensive how the traveling waves contribute to learning. There are several designed equations with arbitrary parameters which their roles in training the network is not clear.

2- As the authors have pointed out in the Discussion, the time scale of learning and that of the traveling wave that comes from neuronal activity are very different. Basically, the variable ψi could reflect any slow variable that relates neuronal activity to one another. Why do the authors call this variable “traveling wave”?

3- How do the authors choose the initial conditions for equation in line 150, when the relation between ψi and the membrane potential is not clear? Is there a global minimum (unique solution for the weights) for any initial conditions?

4- Could the authors plot the input that a target neuron receives as a function of time (line 96), and compare the input that comes from the traveling wave, and the input that comes from the recurrent network dynamics due to spikes? It is important to see the relative contribution of each input component.

5- The roles of the tonic (Dt) and phasic (Dp) Dopamine signal are not clear. How differently do they contribute to learning, what makes the model significantly different that ref 26, and where does the improvement come from? Moreover, the authors used Dt+ Dp for the weight updates (line 126). In none of the equations in the text, Dp or Dt contribute independently/ individually to any other dynamics. Why the authors could not just consider one variable D which has the dynamics τp dD/dt = -D + k.Novelty(t), where k is a constant. The solutions for this equation upon choosing the right k can also give a linear combination of Dt and Dp that show up in line 126. In other words, what the authors have claimed in lines 302 – 306 has never been shown in the main text/ figures/ equations.

6- How is the reward signal integrated in the STDP rule? In line 126, to update the synaptic efficacy, no STDP was used. It is not clear how figure 1B (Hebbian STDP) comes to the picture for synaptic updates.

7- For all the three tasks that the authors have investigated, the mechanisms that are essential for a successful training must be shown in a figure to provide an insight about the key point of the model. Currently, only the success rate as a function of trial numbers, and the reward signals have been shown. The way that weights evolve with time and how they relate to other variables is not clear.

Minor points:

1- “learning distant paths” that shows up in the title is not comprehensive. Do the authors mean strengthening certain connections?

2- It is not clear at all how the traveling waves find the shortest path between neurons. Can the authors explain that in the text (in line 79) how a reward modulated STDP can choose a subset of these paths?

3- Is there any general rule/ insight as to how to change η in the definition of the traveling waves to accommodate for different tasks?

4- In figure 5, there are two different update rules for Dp (one in discrete time the other in continuous time). These two are not consistent. Maybe in the figure caption this has to be explained to avoid confusion.

5- The references to some journals are not consistent in the Reference section (e.g. Journal of Neuroscience,…)

6- Some sentences are not clear in the text. Better and more clear explanations are needed.

7- Figure fonts and resolution have to improve.

**Have all data underlying the figures and results presented in the manuscript been provided?**

Reviewer #1: No: Summary graphs are given in the manuscript along with isolated examples. The underlying data from the multiple simulations that were averaged in the summary graphs were not provided outside of these isolated examples (nor was the generating code provided instead).

Reviewer #2: No: The codes which generate the data are not available.

PLOS authors have the option to publish the peer review history of their article (what does this mean?). If published, this will include your full peer review and any attached files.

Reviewer #1: No

Reviewer #2: No
---

## [Decision Letter · Decision Letter 1]

11 Jan 2021

Dear Mr. Ito,

We greatly apologize for the delay in re-review, and thank you for our patience.  As  New Year's present, we are pleased to inform you that your manuscript 'Learning poly-synaptic paths with traveling waves' has been provisionally accepted for publication in PLOS Computational Biology.

Before your manuscript can be formally accepted you will need to complete some formatting changes, which you will receive in a follow up email. In addition, we recommend that you make your source code available in a public repository. A member of our team will be in touch with a set of requests.  

Best regards,

Kim T. Blackwell, V.M.D., Ph.D.

Deputy Editor

PLOS Computational Biology

Kim Blackwell

Deputy Editor

PLOS Computational Biology

Reviewer's Responses to Questions

**Comments to the Authors:**

Reviewer #1: The authors have addressed all of my concerns and more. The communication of the message and concepts has improved, and the figures are greatly improved, both in terms of content and format. The data and source code has now been included as well. I do suggest a final once-over for the very minor typos that remain.

E.g.:

- lines 345-6: "wave condition successfully establishes"  "the wave conditions successfully establish", "no-wave condition failes"  "the no-wave conditions fail"

- 328: "week"  "weak"

- Fig. 4 caption: "Wave condition (top column) successfully learns correct paths, while no-wave condition (lower column) fails"  same corrections as above, and "column"  "row" in both instances

etc.

I recommend the paper for publication. Cheers.

**Have all data underlying the figures and results presented in the manuscript been provided?**

Reviewer #1: Yes

PLOS authors have the option to publish the peer review history of their article (what does this mean?). If published, this will include your full peer review and any attached files.

Reviewer #1: No

---

## [Editor Report · Acceptance letter]

4 Feb 2021

PCOMPBIOL-D-20-00155R1 

Learning poly-synaptic paths with traveling waves

Dear Dr Ito,

I am pleased to inform you that your manuscript has been formally accepted for publication in PLOS Computational Biology. Your manuscript is now with our production department and you will be notified of the publication date in due course.

With kind regards,

Alice Ellingham
